# Tailored photoenzymatic systems for selective reduction of aliphatic and aromatic nitro compounds fueled by light

Alejandro Prats Luján[1], Mohammad Faizan Bhat[1], Sona Tsaturyan[1], Ronald van Merkerk[1], Haigen Fu[2] & Gerrit J. Poelarends [1]✉

The selective enzymatic reduction of nitroaliphatic and nitroaromatic compounds to aliphatic amines and amino-, azoxy- and azo-aromatics, respectively, remains a persisting challenge for biocatalysis. Here we demonstrate the light-powered, selective photoenzymatic synthesis of aliphatic amines and amino-, azoxy- and azo-aromatics from the corresponding nitro compounds. The nitroreductase from *Bacillus amyloliquefaciens*, in synergy with a photocatalytic system based on chlorophyll, promotes selective conversions of electronically-diverse nitroarenes into a series of aromatic amino, azoxy and azo products with excellent yield (up to 97%). The exploitation of an alternative nitroreductase from *Enterobacter cloacae* enables the tailoring of a photoenzymatic system for the challenging synthesis of aliphatic amines from nitroalkenes and nitroalkanes (up to 90% yield). This photoenzymatic reduction overcomes the competing bio-Nef reaction, typically hindering the complete enzymatic reduction of nitroaliphatics. The results highlight the usefulness of nitroreductases to create selective photoenzymatic systems for the synthesis of precious chemicals, and the effectiveness of chlorophyll as an innocuous photocatalyst, enabling the use of sunlight to drive the photo-biocatalytic reactions.

Amino, azoxy, and azo compounds are extensively used in the dye, electronic, and agrochemical industries[1–3], as well as for the synthesis of numerous blockbuster pharmaceuticals such as Vismodegib (anticancer)[4], Adderall (CNS stimulant)[5], and Phenibut (central nervous system (CNS) depressant)[6]. The synthesis of amino compounds often involves the chemical reduction of a nitro group, employing metal catalysts such as Pd, Zn, SnCl₂, or NiCl₂ (Fig. 1a)[7–11]. The synthesis of azoxy and azo molecules is particularly hazardous, as one of the traditional methodologies, the diazotization, involves the formation of highly reactive diazo intermediates and the use of stoichiometric amounts of nitrite salts and highly corrosive acids[12,13]. Photocatalysis represents a more recent alternative approach for the synthesis of these compounds. This methodology has been used for the synthesis of N-containing molecules via nitro reduction or via amine oxidation, using complex nanoparticles[14–16], transition metals[2,3,17–21], or graphene-based structures (Fig. 1a)[22–25]. Despite the usefulness of these methodologies, there is a requirement for alternate procedures that enable greener and more sustainable synthesis of these highly valuable building blocks.

Enzyme catalysis and photobiocatalysis can represent an attractive alternative for the synthesis of amino, azoxy and azo compounds, especially when applied in the food and pharmaceutical industry[26,27]. Flavoenzymes such as nitroreductases are able to reduce nitro compounds in aqueous conditions with a certain degree of chemo- and regio-selectivity. These enzymes catalyze the reduction of nitro ($-NO_2$) groups into amines ($-NH_2$) via nitroso ($-NO$) and hydroxylamine ($-NHOH$) intermediates by using prosthetic flavin and NAD(P)H cofactors[28–32]. However, the selective reduction of nitro compounds is

¹Department of Chemical and Pharmaceutical Biology, Groningen Research Institute of Pharmacy, University of Groningen, Antonius Deusinglaan 1, 9713 AV Groningen, The Netherlands. ²Department of Chemistry and Chemical Biology, Cornell University, Ithaca, NY 14850, USA. ✉e-mail: g.j.poelarends@rug.nl

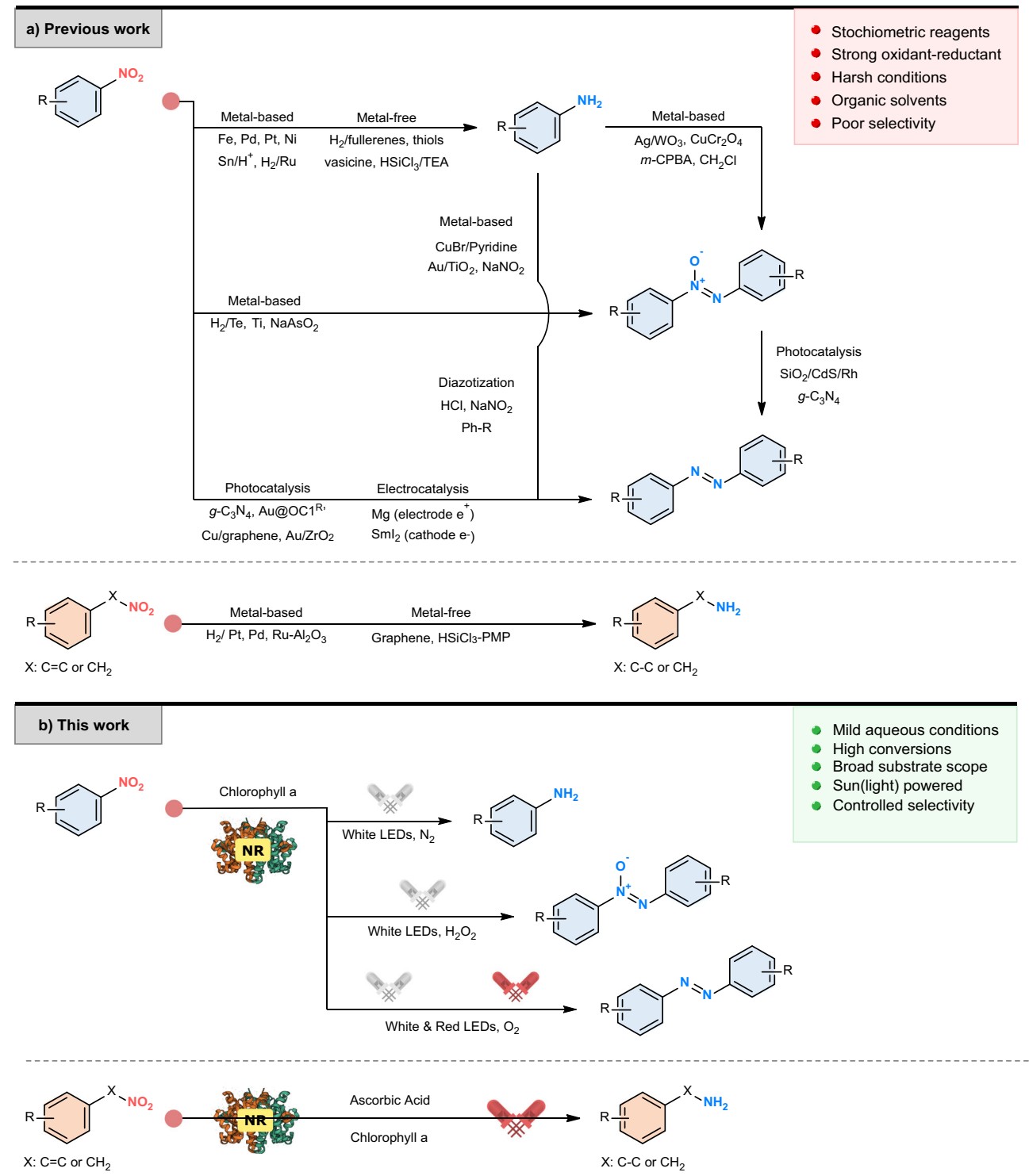

**Fig. 1 | Catalytic reactions of nitro compounds. a** Conventional methodologies and challenges associated with the synthesis of amino, azoxy, and azo compounds. **b** Photoenzymatic model reactions for the synthesis of aliphatic amines and aromatic amino, azoxy, and azo compounds. NR nitroreductase.

a persisting challenge for biocatalysis, frequently resulting in a mixture of intermediates and incomplete reductions[30,33,34]. In addition, the biocatalytic reduction of aliphatic nitro compounds suffers from the so-called bio-Nef reaction, hindering the enzymatic reduction to the amine[35–38]. As such, only a few examples have been reported for amine formation from nitroaliphatics, mostly employing anaerobic microorganisms[39,40]. In recent times, the use of enzymes with a photochemical strategy promoted the emergence of novel methodologies for a variety of redox reactions[41–50]. However, the development of a

photoenzymatic system for the selective reduction of nitro compounds to the desired amino, azoxy, and azo products is, to date, an unmet challenge.

Herein, we report the tailoring of light-fueled photoenzymatic systems, composed of an oxygen-insensitive nitroreductase (either BaNTR1 or EcNR) in synergy with chlorophyll and LEDs light irradiation, for the selective synthesis of aliphatic amines and amino-, azoxy-, and azo-aromatics from the corresponding nitro compounds with high conversions (up to 99%) and excellent yield (up to 97%). The

application of the nitroreductase BaNTR1 promoted the selective photoenzymatic synthesis of amino-, azoxy-, and azo-aromatics from the corresponding nitroarenes, whereas the use of the nitroreductase EcNR enabled the photoenzymatic synthesis of aliphatic amines from the corresponding nitroalkenes and nitroalkanes, overriding the competing bio-Nef reaction (Fig. 1b).

## Results

### Setting up the BaNTR1-based photoenzymatic system

We have previously demonstrated that the flavin-dependent enzyme BaNTR1 was able to perform photoenzymatic ketone reductions in a chemoselective manner[45]. Inspired by these observations, we envisaged that BaNTR1 could potentially be used for the selective photoenzymatic reduction of nitro compounds. We started our investigations by testing whether BaNTR1 can promote the photoenzymatic reduction of nitrobenzene (**1a**, Suppl. Fig. 2), using a variety of photocatalysts under white, blue, or red LEDs light irradiation (Suppl. Fig. 1, Suppl. Table 1). Among the conditions tested, the systems containing chlorophyll with white LEDs and [Ru(bpy)₃]Cl₂ with blue LEDs achieved full conversion of **1a** into aniline (**2**) as a single product (Fig. 2). Afterwards, we decided to perform a series of control experiments in order to determine the role of each component in the reduction and select the optimal combination of LEDs light and photocatalyst (Suppl. Table 2). We observed the conversion of **1a** into **2** (45%) in the absence of nitroreductase and in the presence of blue LEDs, [Ru(bpy)₃]Cl₂, and the cofactor recycling system. This could be the result of the photoexcitation of the NADH cofactor, generated by the cofactor recycling system, under the blue light irradiation, as previously reported[15,45,46,51]. Indeed, no conversion was observed when removing components of the nicotinamide recycling system (glucose,

NAD⁺, or glucose dehydrogenase). In contrast, no background reaction (without BaNTR1) was observed when using chlorophyll as a photocatalyst under white LEDs irradiation. The chlorophyll employed for this study consisted of chlorophyll a (extracted from plants), gum arabic and lactose. Control reactions demonstrated that gum arabic and lactose do not promote the reduction of **1a** into **2** (Suppl. Table 2). Based on these results, we decided to continue our research with the chlorophyll-based photoenzymatic system, as it proved to enhance selective enzymatic reduction without performing additional non-selective photochemical reduction.

### Photoenzymatic synthesis of aromatic amines using BaNTR1

Next, we sought to explore the scope and limitations of this photoenzymatic reaction. Pleasingly, a wide variety of nitroarenes are well accepted by BaNTR1 (Fig. 2, Suppl. Fig. 2, and Suppl. Fig. 3). Nitroarenes possessing electron-withdrawing groups such as halogens, nitrile and trifluoromethyl at different positions are efficiently converted to the desired amine products **3–15** (conversion up to 99%). Notably, only selective reduction of the nitro group to the amine was observed for those substrates containing a carbonyl group (**7** and **15**), highlighting the excellent chemoselectivity of BaNTR1. Substrates containing electron-donating groups such as methoxy, hydroxy, and amine groups can also be fully reduced to the corresponding amine (**16–20**). Furthermore, various dinitrobenzenes and nitroanilines can be fully converted into benzenediamine products (**16, 17, 24–30**, and **31**). More importantly, BaNTR1 can also accept various bulky nitroarenes and nitro-heteroarenes, including the electron-deficient pyridine and electron-rich indole, affording the respective amines (**21–23** and **32–36**) with full conversion. Notably, this photoenzymatic reaction can run on a preparative 0.15 mmol scale and provide the desired amine

**Fig. 2 | Photoenzymatic reduction of nitroarenes into aromatic amines.** Reaction conditions: white LEDs (1000 lx), anaerobic (N₂) 0.1 M MOPS buffer pH = 7 (1ᵃ or 10ᵇ mL), chlorophyll (0.25 mM), nitroarene (5ᵃ or 15ᵇ mM, 10% DMSO), NAD⁺ (0.5 mM), glucose (50 mM), bmGDH (2 μM) and BaNTR1 (10 μM). ᵃThe numbers without brackets denote conversion; the conversion ([product/initial substrate]) was determined by GC–MS (reaction time = 18 h), using toluene as internal standard. ᵇThe numbers within brackets denote isolated yield; the isolated yield (reaction time = 24 h) was calculated after workup and purification. ᶜFrom the corresponding dinitrobenzene substrate (**1f** or **1ab**). ᵈFrom the corresponding nitroaniline substrate. ᵉFrom nitroaniline **1t**. ᶠFrom nitroaniline **1u**. For the corresponding nitro compounds **1a–1aj**, see Suppl. Fig. 2. BaNTR1 nitroreductase from *Bacillus amyloliquefaciens*, bmGDH glucose dehydrogenase from *Bacillus megaterium*.

product in excellent isolated yield (up to 97%) with no changes in chemoselectivity (Fig. 2).

Activity assays with different starting materials and the individual enzyme and photochemical system indicate that particularly the nitroso intermediate is poorly processed by the individual BaNTR1 enzyme, while this intermediate is readily reduced by the full photo-biocatalytic system under the applied conditions (Suppl. Table 8). This suggests that the photocatalytic assistance is likely executed at this step, promoting the synergistic photoenzymatic reduction of the nitroso intermediate to the hydroxylamine, enabling the efficient formation of the desired amine.

## Photoenzymatic synthesis of aromatic azoxy compounds using BaNTR1

After achieving the photoenzymatic synthesis of aromatic amines, we decided to study the selective synthesis of aromatic azoxy compounds by including an oxidant (either $O_2$ or $H_2O_2$) in order to promote the accumulation of nitroso and hydroxylamine intermediates and their spontaneous condensation[52–55]. Satisfyingly, the photoenzymatic system using white LEDs achieved the full and selective conversion of nitrobenzene (1a) into azoxybenzene (37) when employing 15 mM of $H_2O_2$ (Suppl. Fig. 4). Control experiments showed that LEDs light, BaNTR1, chlorophyll, nicotinamide recycling system, and $H_2O_2$ are crucial for the desired full reactivity (Suppl. Table 3 and Suppl. Table 9). Notably, no azoxybenzene product (37) was observed when using aniline as substrate under the optimized reaction conditions, confirming that 37 is not formed from the oxidative pathway (Suppl. Table 3 and Suppl. Table 9)[2,11,19,56,57]. Nitroarene substrates containing electron-withdrawing and electron-donating groups at different

positions are well accepted by BaNTR1, providing the desired aromatic azoxy products 37–51 with excellent conversions and selectivity (Fig. 3 and Suppl. Fig. 4). Remarkably, BaNTR1 is capable of chemoselectively reducing the nitro group over the carbonyl group (48–49). Moreover, the heterocyclic azoxy compound 52 can also be synthesized using this photoenzymatic system. However, starting substrates containing amine, hydroxy, and more complex structures resulted in a mixture of products without the formation of the corresponding azoxy product. In addition, the photoenzymatic system could perform well in semi-preparative-scale reactions, achieving full conversions (>99%) and excellent isolated yields (up to 97%, Fig. 3).

## Photoenzymatic synthesis of aromatic azo compounds using BaNTR1

Inspired by the efficiency and tunability of this photoenzymatic system, we then explored the synthesis of commercially important and synthetically difficult aromatic azo compounds. In our initial studies, we could not observe the reduction of nitrobenzene (1a) into azobenzene (53) with the photoenzymatic system using white LEDs (Suppl. Table 4). On the other hand, the photoenzymatic system composed of chlorophyll and red LEDs achieved the synthesis of azobenzene (53, 47% conversion) when employing a dioxygen-saturated buffer. While the addition of $H_2O_2$ allows the efficient production of the required azoxybenzene (37) intermediate, the subsequent reduction of the azoxybenzene into the corresponding azobenzene (53) is prevented as a consequence of the strong oxidative conditions. As an oxidant is required to efficiently accumulate the azoxy intermediate, we continued our experimentation with the addition of molecular oxygen to the buffer, enabling the synthesis of

**Fig. 3 | Photoenzymatic reduction of nitroarenes into aromatic azoxy and azo compounds.** Azoxy compound reaction conditions: white LEDs (1000 lx), 0.1 M MOPS buffer pH = 7 (1[a] or 10[b] mL), chlorophyll (0.25 mM), nitroarene (5[a] or 15[b] mM, 10% DMSO), NAD⁺ (0.5 mM), glucose (50 mM), $H_2O_2$ (15[a] or 45[b] mM), bmGDH (2 µM) and BaNTR1 (10 µM). [a]The numbers without brackets denote conversion; the conversion was determined by GC-MS (reaction time = 12 h). [b]The numbers within brackets denote isolated yield; the isolated yield was calculated after workup and purification (reaction time = 24 h). Azo compound reaction conditions: white and

red (660 nm) LEDs, dioxygen saturated 0.1 M MOPS buffer pH = 7 (1[a] or 10[b] mL), chlorophyll (0.5 mM), nitroarene (5[a] or 15[b] mM, 10% DMSO), NAD⁺ (0.5 mM), glucose (50 mM), bmGDH (2 µM) and BaNTR1 (10 µM). [a]The numbers without brackets denote conversion; the conversion was determined by GC-MS (total reaction time = 18 h). [b]The numbers within brackets denote isolated yield; the isolated yield (reaction time = 24 h) was calculated after workup and purification. BaNTR1 nitroreductase from *Bacillus amyloliquefaciens*, bmGDH glucose dehydrogenase from *Bacillus megaterium*.

the azoxybenzene intermediate and allowing its further enzymatic reduction into azobenzene **53**. Notably, no azobenzene product was formed when the reaction was performed with aniline (**2**) as the starting material (Suppl. Table 4), indicating that the reaction mechanism follows the nitroarene reduction pathway rather than the aniline oxidation pathway[2,11,19,56,57]. Taking this into consideration and assuming that the prolonged exposure to red LEDs may be detrimental to the selective photoenzymatic reaction, we decided to perform an initial reduction of nitrobenzene (**1**) into azoxybenzene (**37**) using optimal white LEDs, followed by subsequent reduction into azobenzene (**53**) using a shorter period of red LEDs irradiation. To our delight, this tandem photoenzymatic approach (white LEDs followed by red LEDs) achieved the efficient synthesis of a series of azo compounds from corresponding substrates containing electron-withdrawing groups (**53**–**62**) and electron-donating groups (**63**–**64**, Fig. 3 and Suppl. Fig. 5). However, substrates containing amines, carbonyl, hydroxy, or more complex substitutions did not form aromatic azo products. Importantly, the tandem photoenzymatic system was able to maintain the high conversion in the semi-preparative-scale reactions, achieving full conversions (>99%) and excellent yields for all compounds synthesized (up to 97%, Fig. 3). Activity assays with different starting materials and the individual enzyme and photochemical system clearly demonstrate that the enzyme and photocatalyst have to work synergistically to convert the challenging azoxy intermediate, which is not processed by either of the individual catalysts under the applied conditions, into the desired azo compound (Suppl. Table 10).

## Solar-powered photoenzymatic synthesis of aromatic amino and azoxy compounds using BaNTR1

As nature employs chlorophyll to harvest solar energy, we wanted to explore the synthetic usefulness of our photoenzymatic system when fueled by sunlight, an attractive renewable energy source for the synthesis of functionalized aromatics from nitroarenes under mild conditions. A typical scale-up reaction was carried out in a climate-controlled room, under sunlight exposure for 48 h, bearing in mind the variability in sunlight exposure by the periods of night, dawn, and dusk, as well as other meteorological phenomena. The tailored solar-powered photoenzymatic reactions (Fig. 4) resulted in the selective conversion of nitroarenes into the corresponding aromatic amines (**2**, **3**, **12**, **15**, **18**) or azoxy compounds (**37**, **46**, **49**, **50**). The somewhat lower conversion of nitroarenes to anilines compared to the photo-reactor system could be the result of insufficient sunlight exposure from typical Dutch meteorological conditions.

## Photoenzymatic synthesis of aliphatic amines using EcNR

Finally, we explored the challenging photoenzymatic synthesis of aliphatic amines. In contrast to the reduction of nitroarenes, BaNTR1 was not able to perform the reduction of two representative substrates (**65a** and **66a**) selected for the synthesis of aliphatic amines (Fig. 5, Suppl. Fig. 2). By screening an in-house panel of flavoenzymes[45], we found that the nitroreductase EcNR, which also exhibits activity towards nitroaromatics, achieved the formation of minor amounts of the respective aliphatic amine product from both nitro compounds (Suppl. Table 5). During the optimization of the reaction conditions, we observed that the addition of the mild reducing agent, ascorbic acid, improved the conversion into amine to 25–37% (Suppl. Table 6). Alternatively, when the reaction was performed under photo-biocatalytic conditions (chlorophyll and red LEDs), the amine synthesis also showed a remarkable increase in conversion (64–73%). Based on these results, we envisioned that the synergistic combination of the photoenzymatic system and the reducing agent could lead to the effective photobiocatalytic synthesis of aliphatic amines, achieving, in fact, a noteworthy amine formation (**68**, 98% and **78**, 93%). Control reactions that contained NADH or free FMN resulted in minor ene-reduction for substrate **65a**, as previously reported[15,45,51], but no

conversion for substrate **66a** (Suppl. Table 7). After demonstrating the need for EcNR, LEDs light, chlorophyll, and ascorbic acid, we explored the substrate scope of the photoenzymatic reaction with a selection of electronically diverse nitroalkenes and nitroalkanes (Fig. 5, Suppl. Fig. 6). Interestingly, EcNR can process nitroalkenes and nitroalkanes harboring a phenyl ring (or cyclohexene group), which can either be in conjugation with the nitro functionality or separated from the nitro group by a methylene bridge. For the panel of nitroalkenes, the EcNR-based photoenzymatic system performed the reduction of electronically diverse substrates, reducing the nitro group as well as the carbon-carbon double bond, achieving the synthesis of the corresponding aliphatic amines (**68**–**77**, up to >99%). For the set of nitroalkanes, the EcNR-based photoenzymatic system achieved excellent conversions for compounds carrying electron-donating and electron-withdrawing substitutions in *meta*- or *para*- positions (**78**–**82**, up to 97%). Noteworthy, the photoenzymatic reduction of the substrate 1-nitro-1-cyclohexene (**67a**) to produce the cyclohexylamine (**83**) was also accomplished with excellent conversion (93%). In contrast, we observed that *ortho*- and *β*-substituted substrates, as well as those containing hydroxy- and amino-substitutions, were not reduced, perhaps resulting from steric hindrance and/or a deactivating effect (Suppl. Fig. 2, B2). Gratifyingly, the EcNR-based photoenzymatic system maintained excellent substrate conversions in the semi-preparative-scale reactions, obtaining all the aliphatic amine products in high to excellent isolated yield (72–90%, Fig. 5). Additional activity assays indicate that the photochemical system works in synergy with the enzyme, further supported by ascorbic acid, to efficiently reduce the difficult nitroso intermediate into the hydroxylamine, avoiding the undesired tautomerization pathway to the oxime and enabling the effective synthesis of the desired amine (Suppl. Table 11).

## Discussion

We demonstrated that by tailoring an oxygen-insensitive nitroreductase- and chlorophyll-based photoenzymatic system, as well as the redox conditions, the selective photoenzymatic synthesis of aliphatic amines and amino-, azoxy- and azo-aromatics from the corresponding nitro compounds can be achieved. Performing the photoenzymatic reaction under anaerobic conditions allows the selective synthesis of aromatic amines from the corresponding nitroarenes. On the other hand, the addition of a strong oxidant ($H_2O_2$) promotes the photoenzymatic formation of nitroso and hydroxylamine intermediates, followed by their spontaneous condensation into azoxy products. Interestingly, the addition of a milder oxidant (molecular oxygen) to the photoenzymatic reaction leads toward the accumulation of azoxy products while allowing a further reduction into aromatic azo compounds. Finally, the selection of an alternative nitroreductase, EcNR, and the inclusion of a mild reducing agent, ascorbic acid, allows for the efficient photobiocatalytic synthesis of a series of aliphatic amines. These photoenzymatic systems could even be fueled by sunlight, as demonstrated by the selective conversion of nitroarenes into aromatic amine (up to 83%) or azoxy products (>99%). A plausible reaction pathway would follow a sequence of enzyme-mediated hydride transfer and photobiocatalytic radical reduction steps (Suppl. Fig. 8 and corresponding text)[44,45,48–50,58–60]. Given that the nitroso and azoxy intermediates are poorly processed by the individual enzyme system but are readily reduced by the full photobiocatalytic system under the applied reaction conditions (Suppl. Tables 8–11), we propose that the reduction of these challenging intermediates is initiated by photocatalyst-assisted single-electron transfer. The synergistic reduction will be repeated through multiple steps until the final product is formed, depending on the redox environment created by the optimized reaction conditions (Suppl. Fig. 8). Additionally, ascorbic acid could promote the reduction of reaction intermediates during the conversion of nitroaliphatics to the corresponding amines[61,62], hampering the undesired nitroso tautomerization into the oxime

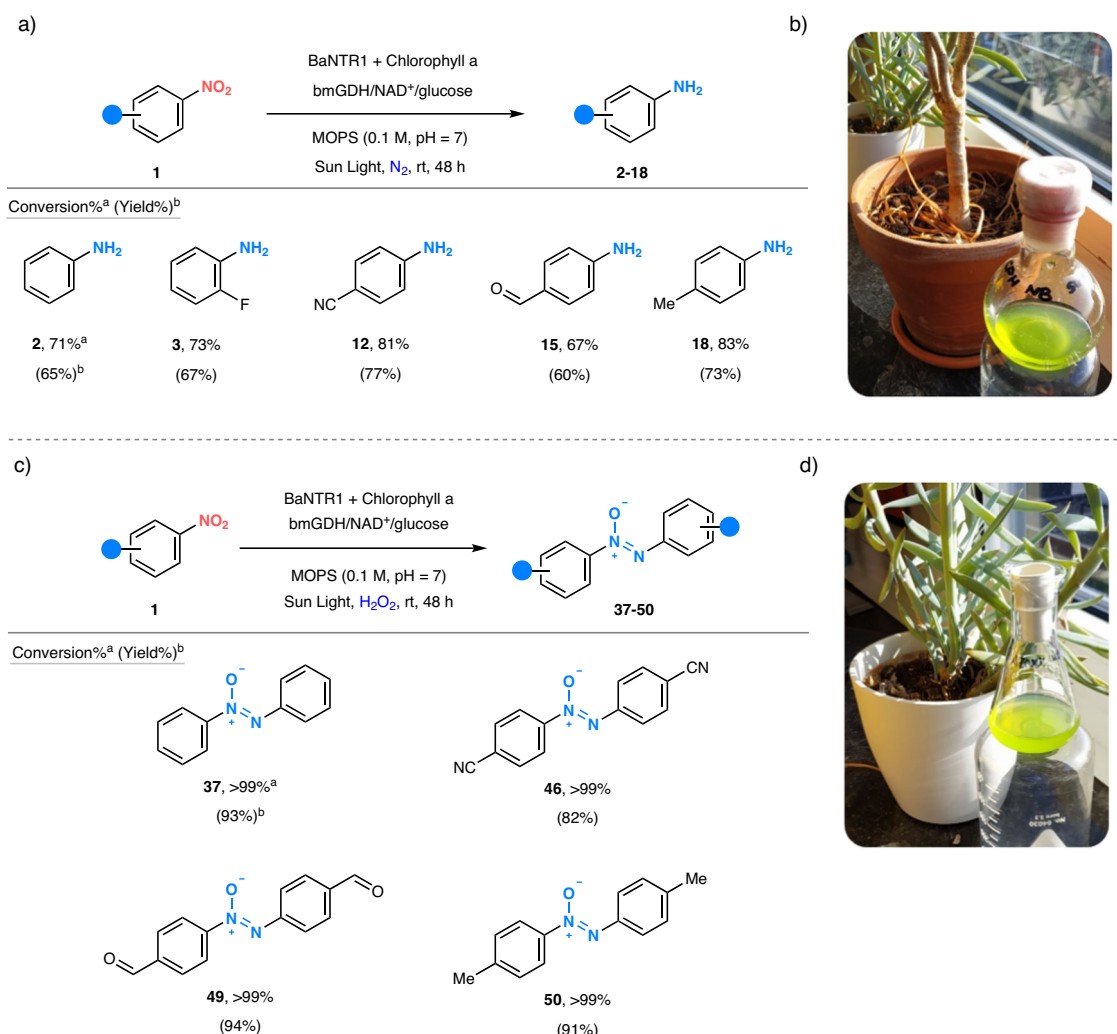

**Fig. 4 | Solar-fueled photoenzymatic reduction of nitroarenes into aromatic amino and azoxy compounds. a** Aromatic amine reaction conditions: sunlight, anaerobic (N₂) 0.1 M MOPS buffer pH = 7 (10 mL), chlorophyll a (0.25 mM), nitroarene (15 mM, 10% DMSO), NAD⁺ (0.5 mM), glucose (50 mM), bmGDH (2 μM) and BaNTR1 (10 μM). **b** Reaction setup for solar-fueled photoenzymatic reduction of nitroarenes into aromatic amines. **c** Azoxy compound reaction conditions: sunlight, 0.1 M MOPS buffer pH = 7 (10 mL), nitroarene (15 mM, 10% DMSO), NAD⁺ (0.5 mM), glucose (50 mM), H₂O₂ (45 mM), bmGDH (2 μM) and BaNTR1 (10 μM). **d** Reaction setup for solar-fueled photoenzymatic reduction of nitroarenes into azoxy-aromatics. [a]The numbers without brackets denote conversion; the conversion was determined by GC–MS (reaction time = 48 h). [b]The numbers within brackets denote isolated yield; the isolated yield (reaction time = 48 h) was calculated after workup and purification. BaNTR1 nitroreductase from *Bacillus amyloliquefaciens*, bmGDH glucose dehydrogenase from *Bacillus megaterium*.

compound and thus assisting the photoenzymatic system to successfully overcome the bio-Nef reaction. We recently reported that nitroreductase BaNTR1 can perform the reduction of ketones following a presumed ketyl radical formation under photoenzymatic conditions[45]. In addition, the reduction of nitroarenes via single electron transfers has been reported previously using light-mediated photochemical radical pathways[25]. The photoenzymatic systems reported herein represent an alternative to chemical and photochemical methods, overcoming the inability to selectively achieve multiple products from a single starting material and effectively producing highly valuable aliphatic amines from the corresponding nitro compounds. Ongoing research aims to provide structural insights into the interactions of the nitroreductase with the non-enzymatic components involved in the photoenzymatic reactions, contributing to a better understanding of the precise reaction mechanisms.

## Methods
### General information
Nitro compounds, chlorophyll a, and [Ru(bpy)₃]Cl₂ were purchased from Sigma-Aldrich Chemical Co. (St. Louis, MO), TCI Europe N.V., or Fluorochem Co. (UK). Solvents were purchased from Biosolve (Valkenswaard, The Netherlands) or Sigma-Aldrich Chemical Co. Ingredients for buffers and media were obtained from Duchefa Biochemie (Haarlem, The Netherlands) or Merck (Darmstadt, Germany). Ni-sepharose 6 fast flow resin and HiLoad 16/600 Superdex 200 pg column were purchased from GE Healthcare Bio-Sciences AB (Uppsala, Sweden). Gas chromatography was performed on a GCMS-QP2010 SE from SHIMADZU®. The column employed was the Agilent J&W HP-5 ms (19091S-433). Running GC method employed: initial 60 °C (hold, 2 min) to 100 °C (5 °C/min) to 300 °C (14 °C/min). Data was analyzed using the software LabSolutions™ post-run analysis from SHIMADZU®. NMR analysis was performed on a Brucker 500 MHz machine at the drug design laboratory of the University of Groningen (RUG). ¹H NMR data are reported as follows: chemical shift (δ ppm), multiplicity (s = singlet, brs = broad singlet, d = doublet, t = triplet, q = quartet, m = multiplet, dd = doublet of doublet, dt = doublet of triplet, ddd = doublet of doublet of doublet), coupling constant (Hz), and integration. Conversion and yield were determined via [product]/[initial substrate].

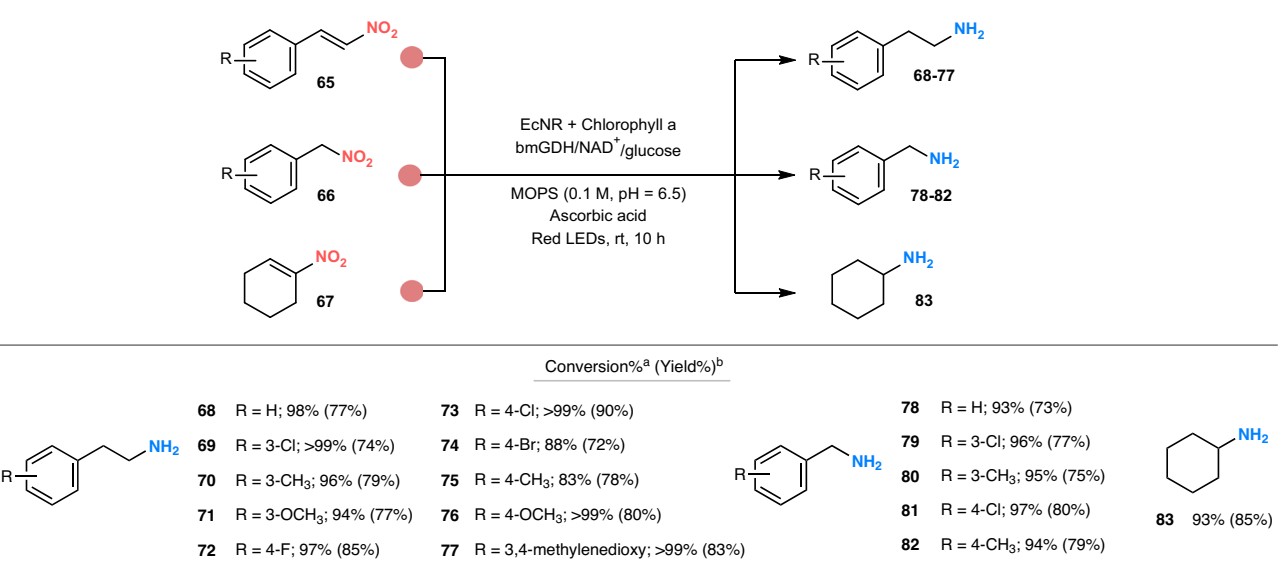

**Fig. 5 | Photoenzymatic reduction of nitroaliphatic compounds into aliphatic amines.** Reaction conditions: red LEDs (660 nm, 10[a] or 16[b]), 0.1 M MOPS buffer pH = 6.5 (1[a] or 10[b] mL), chlorophyll a (0.5 mM), substrate (5[a] or 15[b] mM, 10% DMSO), NAD+ (0.5 mM), glucose (50 mM), ascorbic acid (20[a] or 60[b] mM), bmGDH (2 μM) and EcNR (20 μM). [a]The numbers without brackets denote conversion; the conversion was determined by GC–MS (reaction time = 10 h). [b]The numbers within brackets denote isolated yield; the isolated yield (reaction time = 16 h) was calculated after workup and purification. For the corresponding nitro compounds (**65a–65j**, **66a–66e** and **67a**), see Suppl. Fig. 2. EcNR nitroreductase from *Enterobacter cloacae*, bmGDH glucose dehydrogenase from *Bacillus megaterium*.

### Enzyme expression and purification

His-tagged BaNTR1, EcNR, and bmGDH proteins were expressed in *Escherichia coli* BL21(DE3), employing pET21a as an expression vector[45]. Freshly transformed BL21(DE3) cells were grown at 37 °C in 1 L of 2xYT medium containing ampicillin (100 μg/ml). At an $OD_{600}$ of 0.6–0.8, isopropyl β-D-1-thiogalactopyranoside (0.3 mM) was added, and the culture was grown overnight at 18 °C. Cells were pelleted and resuspended in lysis buffer (50 mM Tris-HCl pH 8; 300 mM NaCl; 10 mM imidazole). Next, cells were sonicated and centrifuged at 12,000*g* at 4 °C for 30 min to remove cell debris and obtain cell-free extract. The cell-free extract was incubated with Ni–NTA agarose beads, beforehand equilibrated with lysis buffer at 4 °C for 3 h under continuous rotation. The mixture was loaded into a gravity flow column and washed with 1 column volume of lysis buffer and 2 column volumes of washing buffer (50 mM Tris-HCl pH = 8; 300 mM NaCl; 30 mM imidazole). Retained proteins were eluted with 5 mL of elution buffer (50 mM Tris-HCl pH = 8; 300 mM NaCl; 500 mM imidazole). To restore the purified enzyme with the FMN prosthetic group, purified enzymes were incubated with free FMN (1 mM) for 3 h at 4 °C under constant rotation. Finally, enzymes were desalted and switched to the appropriate buffer using PD-10 desalting columns (GE Healthcare). Enzymes were flash-frozen using liquid nitrogen and stored at −20 °C or −80 °C until further use.

### Analytical scale photoenzymatic reactions for GC–MS analysis

Reactions for the photoenzymatic synthesis of aromatic amines (**2**–**36**, Suppl. Figs. 9–44) were performed overnight (18 h) in a white LEDs photoreactor at room temperature (cooled by air to keep a temperature of 25 (±3) °C in the photoreactor), placed in a circular disposition at a distance of 6 cm from the light source, using a capped glass vial containing: 0.1 M anaerobic (N2) MOPS buffer pH = 7 (1 mL), chlorophyll a (0.25 mM), nitroarene (5 mM, 10% DMSO), NAD+ (0.5 mM), glucose (50 mM), bmGDH (2 μM) and BaNTR1 (10 μM, see Suppl. Fig. 7 for a UV–vis spectrum of a representative reaction mixture).

Inorganic buffers (NaPi, NaHCO3, and KH2PO4) can also support the photobiocatalytic process, albeit resulting in somewhat lower conversion (62–77% conversion of nitrobenzene into aniline) compared to the transformation in MOPS (99% conversion of nitrobenzene into aniline).

Reactions for the photoenzymatic synthesis of aromatic azoxy compounds (**37**–**52**) were performed for 12 h in a white LEDs photoreactor at room temperature (cooled by air to keep a temperature of 25 (±3) °C in the photoreactor), placed in a circular disposition at a distance of 6 cm from the light source, using a capped glass vial containing: 0.1 M MOPS buffer pH = 7 (1 mL), chlorophyll a (0.25 mM), nitroarene (5 mM, 10% DMSO), NAD+ (0.5 mM), glucose (50 mM), H2O2 (15 mM), bmGDH (2 μM), and BaNTR1 (10 μM).

Reactions for the photoenzymatic synthesis of aromatic azo compounds (**53**–**64**) were performed for 12 h in a white LEDs photoreactor followed by an additional 6 h in a red LEDs (660 nm) photoreactor at room temperature (cooled by air to keep a temperature of 25 (±3) °C in the photoreactor), placed in a circular disposition at a distance of 6 cm from the light source, using a capped glass vial containing: dioxygen saturated 0.1 M MOPS buffer pH = 7 (1 mL), chlorophyll a (0.5 mM), nitroarene (5 mM, 10% DMSO), NAD+ (0.5 mM), glucose (50 mM), bmGDH (2 μM), and BaNTR1 (10 μM). Additional experiments with a constant inflow of molecular oxygen into the reaction mixture using an oxygen-filled balloon were also performed, but these conditions did not significantly improve the reaction rate or the total conversion. Given that red light is part of the spectrum of white light, the use of increased reaction time could potentially also result in the photoenzymatic conversion of nitrobenzenes into azobenzenes without the need for a switch from white to red light. However, we have chosen not to use prolonged incubation times to avoid possible issues with photobleaching and inefficient cooling.

Photobiocatalytic reactions for the synthesis of aliphatic amines (**68**–**83**, Suppl. Figs. 45–60) were performed for 10 h in a red LEDs (660 nm) photoreactor at room temperature (cooled by air to keep a temperature of 25 (±3) °C in the photoreactor), placed in a circular disposition at a distance of 6 cm from the light source, using a capped glass vial containing: 0.1 M MOPS buffer pH = 6.5 (1 mL), chlorophyll a (0.5 mM), substrate (5 mM, 10% DMSO), NAD+ (0.5 mM), ascorbic acid (20 mM), glucose (50 mM), bmGDH (2 μM), and EcNR (20 μM).

The photoreactors employed were designed in-house, using white, blue (λ max 440 nm), or red (λ max 660 nm) LED stripes (19 W, 1000 lx) coupled with air refrigeration. Chlorophyll employed was obtained from TCI Chemicals (C0780), with a lambda max at 660–670 nm (diethyl ether). All reaction products were extracted with an equivalent volume (1:1) of ethyl acetate three times, and 5 µL of the organic solvent was injected into the GC–MS for product analysis (using toluene as the internal standard).

## Semi-preparative scale photobiocatalytic reactions for NMR and yield analysis

The photoenzymatic semi-preparative scale reactions for aromatic amine synthesis (2–36, Suppl. Figs. 61–85) were performed for 24 h in the same white LEDs photoreactor as employed for analytical scale reactions, at room temperature and using a glass vial containing: anaerobic ($N_2$) 0.1 M MOPS buffer pH = 7 (10 mL), chlorophyll a (0.25 mM), nitroarene (15 mM, 10% DMSO), $NAD^+$ (0.5 mM), glucose (50 mM), bmGDH (2 µM, glucose dehydrogenase), and BaNTR1 (10 µM).

Photoenzymatic semi-preparative scale reactions toward aromatic azoxy synthesis (37–52, Suppl. Figs. 86–98) were performed for 24 h in the same white LEDs photoreactor as employed for analytical scale reactions at room temperature and using a glass vial containing: 0.1 M MOPS buffer pH = 7 (10 mL), chlorophyll a (0.25 mM), nitroarene (15 mM, 10% DMSO), $H_2O_2$ (45 mM), $NAD^+$ (0.5 mM), glucose (50 mM), bmGDH (2 µM, glucose dehydrogenase), and BaNTR1 (10 µM).

Photoenzymatic semi-preparative scale reactions toward aromatic azo synthesis (53–64, Suppl. Figs. 99–111) were performed for 24 h in the same white LED photoreactor plus another 8 h in the same red LEDs photoreactor as employed for analytical scale reactions, at room temperature and using a glass vial containing: dioxygen saturated ($O_2$) 0.1 M MOPS buffer pH = 7 (10 mL), chlorophyll a (0.5 mM), nitroarene (15 mM, 10% DMSO), $NAD^+$ (0.5 mM), glucose (50 mM), bmGDH (2 µM, glucose dehydrogenase), and BaNTR1 (10 µM).

Photobiocatalytic reactions for the synthesis of aliphatic amines (68–83, Suppl. Figs. 112–130) were performed for 16 h in the same red LEDs (660 nm) photoreactor employed for analytical scale at room temperature and using an Erlenmeyer flask containing: 0.1 M MOPS buffer pH = 6.5 (10 mL), chlorophyll a (0.5 mM), substrate (15 mM, 10% DMSO), $NAD^+$ (0.5 mM), glucose (50 mM), ascorbic acid (60 mM), bmGDH (2 µM, glucose dehydrogenase), and EcNR (20 µM).

Products from the photoenzymatic reactions were extracted with an equivalent volume (1:1) of ethyl acetate three times, the solvent dried with $Na_2SO_4$ and then removed in vacuo to give crude product. Then, photoenzymatic products were purified with normal silica-gel chromatography (amine and azoxy products) and semi-preparative TLC (azo products) from 10% EtOAc/Pentane to 50% EtOAc/Pentane. In all cases, the organic solvent was then evaporated at room temperature under vacuum. The isolated yield was calculated at this point, and the respective product was resuspended in the appropriate NMR solvent: DMSO$_{d6}$ or Chloroform-d. Some characterized products showed chemical shifts and coupling constants slightly different from previously reported data, which appears to be the result of the solvent influence and variances in the setup of the NMR machines.

## Solar-powered photoenzymatic synthesis of aromatic amino and azoxy compounds

Solar-powered photoenzymatic synthesis of aromatic amines (2–18) was performed in a climate-controlled room for 48 h under sun exposure, bearing in mind the variability in sunlight exposure produced by the periods of night, dawn, and dusk, as well as other meteorological phenomena. Reactions were performed during the weekend of March 5th–6th in Groningen, The Netherlands (53°13'25.9" north latitude, 6°34'20.1" east longitude; 35 m above sea level) with a reported average sun intensity of 423 W/m² (https://www.buienradar.nl/weer/Groningen/NL/2755251). The reaction was performed in a

round flask at room temperature and contained: anaerobic ($N_2$) 0.1 M MOPS buffer pH = 7 (10 mL), chlorophyll a (0.25 mM), nitroarene (15 mM, 10% DMSO), $NAD^+$ (0.5 mM), glucose (50 mM), bmGDH (2 µM, glucose dehydrogenase), and BaNTR1 (10 µM). Solar-powered photoenzymatic synthesis of aromatic azoxy compounds (37–50) was performed in a climate-controlled room for 48 h under sun exposure, bearing in mind the variability in sunlight exposure produced by the periods of night, dawn and dusk, as well as other meteorological phenomena. Reactions were performed during the weekdays of March 22nd–23rd in Groningen, The Netherlands (53°13'25.9" north latitude, 6°34'20.1" east longitude; 35 m above sea level) with a reported average sun intensity of 387 W/m² (https://www.buienradar.nl/weer/Groningen/NL/2755251). The reaction was performed in an Erlenmeyer flask at room temperature and contained: 0.1 M MOPS buffer pH = 7 (10 mL), chlorophyll a (0.25 mM), nitroarene (15 mM, 10% DMSO), $H_2O_2$ (45 mM), $NAD^+$ (0.5 mM), glucose (50 mM), bmGDH (2 µM, glucose dehydrogenase), and BaNTR1 (10 µM).

## Data availability

The data generated in this study are provided in the manuscript and in the Supplementary Information.

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

## Acknowledgements

We acknowledge financial support from The Netherlands Organization of Scientific Research (VICI grant 724.016.002 to G.J.P.) and from the European Union's Horizon 2020 research and innovation program under the Marie Skłodowska-Curie grant agreement No. 754425 (to G.J.P.).

## Author contributions

A.P.L. performed most biochemical and (photo)biocatalytic experiments and wrote the first draft of the paper. M.F.B. contributed to the purification of the compounds, performed most of the NMR experiments, analyzed the NMR data, and wrote the corresponding parts of the paper. S.T. contributed to experimental work, and R.v.M. assisted with the design of the photoreactor and with the optimization of the reaction conditions. H.F. assisted in the design of the experiments and edited the paper. G.J.P. initiated the project and supervised the scientific work.

## Competing interests

The authors declare no competing interests.
