## [Peer Review File · Nature Communications]

REVIEWER COMMENTS

Reviewer #1 (Remarks to the Author):

Poelarends and colleagues designed a light-powered selective photoenzymatic method for the synthesis of aliphatic amines and amino-, azoxy- and azo- aromatics from the corresponding nitro compounds. Specifically, via the reaction engineering and along with the combination between chlorophyll and nitroreductase BaNTR1, a variety of nitroarenes was converted into aromatic amino, azoxy and azo products. As the aliphatic nitro compounds are more inert, the authors attempted the second nitroreductase EcNR to perform the reduction reaction to give aliphatic amines under photobiocatalytic conditions. This approach allows a very broad substrate scope, which was not only demonstrated in a preparative scale synthesis but also showcased under sunlight conditions. I do appreciate the great effort to come up with this nice manuscript. The nitro reductions have been well-established in chemical industry with very cheap stoichiometric reductants or chemical catalysts. Though under the anaerobic conditions BaNTR1 showed kind of new chemistry, the reduction reactions using nitroreductases BaNTR1 and EcNR have been well-documented in the literatures, the same is true for chlorophyll promoted light-driven reactions. As such, the reviewer believes that the manuscript does not show a high degree of conceptual novelty, which is certainly required by Nature Communications. Another inadequacy is that the mechanism is still not clear. For example, where did the two protons come from under the anaerobic reduction conditions? Also, the synergic effect between the photocatalyst and the enzymes is not clear. Additionally, the photoenzymatic approach showed a very strong background reaction, e.g. in Table S1, under the conditions without the photocatalyst or in dark the reactions gave 45-48% conversion, which raised the question whether the combination of the photocatalyst is necessary. What would happen if the authors doubled the enzyme concentration, can it lead to complete conversion without the photocatalyst? Overall, I found it a nice work but believe that it belongs to a synthetic journal. The results presented here do not show a significant advance in the field of nitroreduction, neither in the field of (photo)biocatalysis. Hopefully, the authors would find the following comments useful to improve the manuscript.

1. "All control reactions including these components resulted in no conversion." what exactly was this sentence referring to?
2. The authors mentioned "oxygenated buffer", it sounds confusion. Was the buffer oxidized, or simply saturated with dioxygen?
3. The reaction conditions about the amine synthesis need further clarification. Where is the isolated yield, inside the bracket? How was the conversion/yield determined exactly, by $[\text{product}]/[\text{initial substrate}]$, or $[\text{product}]/([\text{substrate}]+[\text{product}] \text{ in real time})$?
4. The note [e] of Figure 2, molecule 1f has no F atom, please check and revise.
5. The note [e] of Figure 3, "Azoxy compound reaction conditions: [a] Conversion determined by GC-MS (reaction time = 12 h). [b] Yield determined by NMR (reaction time = 24 h)". However, the illumination time of the white and red LEDs was not specified respectively for this tandem cascade.
6. In the paragraph of "Photoenzymatic synthesis of aromatic azo compounds using BaNTR1", "Aromatic azo compounds (53-64) were synthesis under oxygenated 0.1 M MOPS buffer pH=7...", Did the author do an experiment to supply extra oxygen during the experiment? Does the extra oxygen improve the conversion in larger scale? It needs to be classified.
7. In the part of "Solar powered photoenzymatic synthesis of aromatic amino and azoxy compounds using BaNTR1", it did not match the supporting information, please revise it. In this part, the reaction time is 48 h, how to preserve the reaction when it was dark?
8. the format of reference "...Science (80-.). 322, 1661-1664 (2008 is incorrect. Please also pay attention to other formatting issues with the references.
9. The format of the reaction process is inconsistent, i.e. chlorophyll a (0.25 mM) on page 4, and 0.25 mM chlorophyll a on page 42
10. It looks as if the reaction temperature was not specified in the manuscript. Temperature has a great influence on the catalytic activity of enzymes. Whether the authors have investigated the effect of nitroreductase on the reduction of nitro substrates at different temperatures? Also, how did the authors control the reaction temperature in the solar-driven experiment?
11. In the SI, some compounds have ^{13}C -NMR spectra while some not, please unify the way of supplementing the data.
12. In Table S1, what was the condition for the standard enzymatic reaction? Only without LEDs light? The same confusion raised in standard aerobic enzymatic reaction of Table S2.
13. Was the light intensity of white, blue and red the same?

14. Double peaks appeared in the gas chromatography in Figure S31 and S38, are the authors sure that this represents a single compound? My experience with GC/HPLC is that this indicated two compounds.
15. By using different starting materials, the reaction mechanism can be proposed and schematically presented.
16. In page 8 the authors claimed that “we could not observe the reduction of nitrobenzene (1a) into azobenzene (53) with the photoenzymatic system using white LEDs (Suppl. Table S4), We hypothesize this to be the result of insufficient (non-optimal) irradiation of the chlorophyll by the white LEDs”, however, in a latter study the product formation was observed under red LED. To me this is strange as the while LED is often composed of red, green and blue emissions.
17. Though in all cases the conversion and yield were reported, the product concentration should be mentioned to see the robustness, as in a photocatalytic setup (especially when O₂ was supplied), evaporation of the substrate and product could happen.
18. It would be very helpful if a time course could be provided, in such a way the readers could see the kinetic of the reaction process.
19. The SI figures S76, S83, S86 and S88, the proton spectra are not clean, and it is recommended that the authors replace them with clean spectra.

Reviewer #2 (Remarks to the Author):

Summary:

The authors set out to fill a gap in the ene reductase (ER)-catalyzed reduction of ene and nitro functionalities: the reduction of aromatic nitro compounds to amines. While ene functionalities and aliphatic nitro compounds have been successfully reduced with photo-assisted ER catalysis, this feat had not been achieved with aromatic nitro compounds. However, while the authors cited Todd Hyster's work (now at Cornell U.), albeit sparsely (a single ref. 42), they did not include Hysrer's most recent work, nor the important recent work by Bornadel et al. (*Org. Process Res. Dev.* 2021, 25, 648–653) and Bisagni et al. (*Curr Res Chem Biol*, 2022, 100026) from Johnson Matthey and Amgen. This has to be fixed in the revision.

That issues aside, the presented work has been performed with care and the results are interesting. The finding of very high chemoselectivity dependent on the oxidant (to azo or azoxy compounds) or the presence of an inert atmosphere (to amines) is highly significant. Likewise for overriding the Nef reaction in the case of aliphatic nitro compounds.

After revision, this manuscript should be considered for acceptance.

Detailed critique:

Figures 2, 3, and ff: In the Figure legend, clarify the meaning of the numbers next to each product, both before and inside the parentheses: is this conversion and isolated yield?

Except for the aliphatic amines, the manuscript uses EcNR and BaNTR1 for very different reactions. Is there no cross-reactivity? If so, the reactivity of the 'other' enzyme should be clearly explained; if not, it should be clearly mentioned.

How did other enzymes on the 'in-house' panel (which enzymes does that entail?) do on the aromatic substrates? The manuscript seems to carefully and selectively report positive results (which is great) but should also mention, within the scope, what did not work.

Transformation of 67 to 83: does the nitrocyclohexane also react? How? Selectively? If not, why not?

Reviewer #3 (Remarks to the Author):

The authors developed a combined photochemical and biocatalytic approach to convert nitro compounds into amines, azoxy- and azo- compounds. The intriguing feature of the study is that the desired reaction route can be selected, based on the reaction conditions. The authors demonstrate the versatility of their system on large library of electronically and structurally distinct aromatic and aliphatic substrates. Unfortunately, no additional information besides the extended substrate scope is provided. From my perspective the reaction mechanism of the distinct reactions (enzymatic and photochemical) should be elucidated. As the reduction proceeds via a range of intermediates (e.g., nitroso, hydroxylamine, imine, oxime), these intermediates should be characterized.

Overall, the study provides exciting results, but many questions remain open:

-The demonstrated systems work to some degree also without the use of light and also without the use of enzyme:

oThe reduction to the amine works with the ruthenium based photocatalyst without enzyme. I wonder whether chlorophyll and red light, or using chlorophyll with a higher light intensity would also show some reactivity. The reaction works also without the photocatalyst. This suggests to me that the photocatalyst and the enzyme do have different reaction rates for the different reduction intermediates and in concert both can play their strengths. However, in order to elucidate this, the controls need to be made with the potential intermediates (nitroso, hydroxylamine, imine, oxime) using the full system, only the enzymatic system and only the photochemical system. This should provide more data on the individual catalysts` contribution to the overall reaction.

oThe formation of the azoxy-compounds would work without applying the photochemical system. Similar as in the previous point, the reactivities of the intermediates should be characterized.

oThe conversion of the aliphatic compounds requires the addition of ascorbate, which again, on its own would perform the reaction. Again, the two catalytic systems should be tested with all potential reaction intermediates.

-All this information should be collected in order to propose a reaction sequence with the individual intermediates. Only then the role of the photochemical and the biocatalyst is clear.

-The formation of the azoxy-compounds requires a switch of light. This is surprising, as red light should be part of the spectrum of the applied white light. Is the reason the different light intensity? Then either increased reaction time or increased intensity of white light should give the same effect. This effect should be explained.

-The reaction conditions of the photochemical reactions are not sufficiently documented. What are the different applied light intensities and emission spectra? This data should be available from the manufacturer of the LED. It is my understanding that the authors utilized a custom illumination system. In this case the distance between sample and light source should be provided. Also, the light intensity in the reactor should be put into context with the light intensity that was available in the reported solar-powered reactions. Finally, the absorption spectrum of the reaction mixture and the photocatalyst should be provided in order to discuss the requirement of two different light sources for the formation of the azo compounds.

-The contribution of the different electron donors should be unraveled by control experiments. The applied buffer compound MOPS was demonstrated to serve as electron donor in photochemistry. In addition, a glucose based NADH regeneration system is applied. MOPS might promote the photochemical reaction, glucose the biocatalytic one. Alternate buffer salts and water should be tested as solvents. The number of NADH equivalents that are utilized in the reaction should be measured and correlated to the number of electrons that are required for the reduction. And the two processes (photochemical and biocatalytic) should be tested individually with the different possible electron donors.

-Page 6: Figure 2 and the following line: "Furthermore, various dinitrobenzenes and nitroanilines can be fully converted into the benzenediamine products (24-30 and 31)." Substrates 16 and 17 also belong to this group.

-Caption of figure 2: I believe in point [e] the wrong substrate number is given. Maybe 1g instead?

-The captions of all figures: the meaning of [a] and [b] is not obvious. I suggest writing "numbers in brackets" and "numbers without brackets"

-Table S6: reactions performed on "bench": does this mean without light? This should be clarified in the text

-It would be great to see an aliphatic substrate that might give an e.e. to demonstrate the enzyme's stereoselectivity

Response to the reviewers' comments

We appreciate the constructive remarks of the reviewers as well as the decision to consider a revised manuscript, which addresses the concerns. The reviewer's comments are provided below along with our responses (in italics). The changes in the text are highlighted by a yellow background color in the revised manuscript and Supplementary Information.

Reviewer #1 (Remarks to the Author):

Poelarends and colleagues designed a light-powered selective photoenzymatic method for the synthesis of aliphatic amines and amino-, azoxy- and azo- aromatics from the corresponding nitro compounds. Specifically, via the reaction engineering and along with the combination between chlorophyll and nitroreductase BaNTR1, a variety of nitroarenes was converted into aromatic amino, azoxy and azo products. As the aliphatic nitro compounds are more inert, the authors attempted the second nitroreductase EcNR to perform the reduction reaction to give aliphatic amines under photobiocatalytic conditions. This approach allows a very broad substrate scope, which was not only demonstrated in a preparative scale synthesis but also showcased under sunlight conditions. I do appreciate the great effort to come up with this nice manuscript. The nitro reductions have been well-established in chemical industry with very cheap stoichiometric reductants or chemical catalysts. Though under the anaerobic conditions BaNTR1 showed kind of new chemistry, the reduction reactions using nitroreductases BaNTR1 and EcNR have been well-documented in the literatures, the same is true for chlorophyll promoted light-driven reactions. As such, the reviewer believes that the manuscript does not show a high degree of conceptual novelty, which is certainly required by Nature Communications.

Reply: The selective reduction of nitro compounds into the desired product (amine, azoxy or azo compound) is a fundamental challenge for biocatalysis, normally resulting in a mixture of intermediates and incomplete reductions. As underlined by the other two reviewers, the tailoring of photoenzymatic systems that enable the highly selective synthesis of amine, azoxy or azo products from nitro compounds is highly significant. Likewise, the development of a photoenzymatic system that overcomes the competing 'bio-Nef' reaction, enabling the efficient synthesis of aliphatic amines from aliphatic nitro compounds is very important. We therefore strongly believe our study has a high degree of conceptual novelty and would be well received by the scientific community.

Another inadequacy is that the mechanism is still not clear. For example, where did the two protons come from under the anaerobic reduction conditions? Also, the synergic effect between the photocatalyst and the enzymes is not clear. Additionally, the photoenzymatic approach showed a very strong background reaction, e.g. in Table S1, under the conditions without the photocatalyst or in dark the reactions gave 45-48% conversion, which raised the question whether the combination of the photocatalyst is necessary. What would happen if the authors doubled the enzyme concentration, can it lead to complete conversion without the photocatalyst? Overall, I found it a nice work but believe that it belongs to a synthetic journal.

Reply: The selective synthesis of amine, azoxy or azo products from nitro compounds is not possible with only the enzyme, which produces a mixture of different products, or only the photocatalyst, which under the applied reaction conditions cannot independently reduce the substrate or any of the potential intermediates. To further clarify the individual contribution of the enzymatic and photochemical system to the overall reaction, we have exposed substrate and observed reduction intermediates or products to the different catalytic systems (only the enzymatic system, only the photochemical system, and the full photobiocatalytic system). We have examined the resulting products by GC-MS; the results have been included as Tables S8-S11 in the revised Supplementary Information. It is clear that the enzyme is not very selective and generates multiple products, mainly because of inadequate reduction of the nitroso (or azoxy) intermediate. The outcomes also indicate that the photochemical system (chlorophyll) cannot perform the reduction of any substrate or intermediate by itself, but apparently works in concert with the enzyme to reduce the challenging nitroso and azoxy intermediates (of course depending on the applied reaction conditions). Indeed, the nitroso and azoxy compounds are poorly processed by the individual enzyme system, but are readily converted by the full photobiocatalytic system under the applied reaction conditions. Notably, the individual enzyme can process the nitro and hydroxylamine compounds quite well. Based on these new and previous results, a mechanism has been proposed for the reduction pathway (Scheme S1 and corresponding text in the revised SI). The Discussion section of the main text has been rephrased accordingly. We feel that we have appropriately addressed the most important concerns, with more detailed mechanistic aspects being beyond the scope of this study.

The results presented here do not show a significant advance in the field of nitroreduction, neither in the field of (photo)biocatalysis. Hopefully, the authors would find the following comments useful to improve the manuscript.

1. “All control reactions including these components resulted in no conversion.” what exactly was this sentence referring to?

*Reply: As this question might also occur to a reader, we have rephrased this sentence. It now reads “Control reactions demonstrated that gum arabic and lactose do not promote the reduction of **1a** into **2** (Suppl. Table S2).”*

2. The authors mentioned “oxygenated buffer”, it sounds confusion. Was the buffer oxidized, or simply saturated with dioxygen?

Reply: The buffer was saturated with dioxygen. This is now clearly mentioned in the revised manuscript.

3. The reaction conditions about the amine synthesis need further clarification. Where is the isolated yield, inside the bracket? How was the conversion/yield determined exactly, by [product]/[initial substrate], or [product]/([substrate]+[product] in real time)?

Reply: Indeed, the isolated yield is given inside the brackets. For clarity, we have rephrased the corresponding sentence in the legend of Figure 2, which now reads: “[a] The numbers without brackets denote conversion; the conversion ([product/initial substrate]) was determined by GC-MS (reaction time = 18 h), using toluene as internal standard. [b] The numbers within brackets denote isolated yield; the isolated yield (reaction time = 24 h) was calculated after workup and purification.

4. The note [e] of Figure 2, molecule 1f has no F atom, please check and revise.

Reply: The reviewer is correct. Accordingly, we have adjusted this in the revised manuscript.

5. The note [e] of Figure 3, “Azoxy compound reaction conditions: [a] Conversion determined by GC-MS (reaction time = 12 h). [b] Yield determined by NMR (reaction time = 24 h)”. However, the illumination time of the white and red LEDs was not specified respectively for this tandem cascade.

Reply: Although these illumination times were already specified in the Supplementary Information, it is an excellent suggestion to also include this information in the main text. In the revised manuscript, we therefore have now included the illumination times in the scheme at the top of Figure 3.

6. In the paragraph of “Photoenzymatic synthesis of aromatic azo compounds using BaNTR1”, “Aromatic azo compounds (53-64) were synthesis under oxygenated 0.1 M MOPS buffer pH=7...”, Did the author do an experiment to supply extra oxygen during the experiment? Does the extra oxygen improve the conversion in larger scale? It needs to be classified.

Reply: This is an excellent suggestion. Accordingly, we have performed additional experiments with a constant inflow of molecular oxygen into the reaction mixture using an oxygen-filled balloon, but these conditions did not significantly improve the rate nor the total conversion for either the analytical scale or the semi-preparative scale reactions. This is now also mentioned in the respective section of the revised Supplementary Information.

7. In the part of “Solar powered photoenzymatic synthesis of aromatic amino and azoxy compounds using BaNTR1”, it did not match the Supplementary Information, please revise it. In this part, the reaction time is 48 h, how to preserve the reaction when it was dark?

Reply: The reviewer raises an important point that might also occur to a reader. We therefore have rephrased the corresponding sentence in the revised manuscript, which now reads: “A typical scale-up reaction was carried out under sunlight exposure for 48 h, bearing in mind the variability in light intensity produced by the periods of night, dawn and dusk, as well as other meteorological phenomena.” This sentence has also been added to the respective section of the Supplementary Information. As requested, the format of the experimental part of the Supplementary Information has been revised to match the manuscript format.

8. the format of reference "...Science (80-.). 322, 1661–1664 (2008 is incorrect. Please also pay attention to other formatting issues with the references.

Reply: The format of the references has been checked and corrected where needed.

9. The format of the reaction process is inconsistent, i.e. chlorophyll a (0.25 mM) on page 4, and 0.25 mM chlorophyll a on page 42.

Reply: This has been adjusted. The format is now consistent in the revised Supplementary Information.

10. It looks as if the reaction temperature was not specified in the manuscript. Temperature has a great influence on the catalytic activity of enzymes. Whether the authors have investigated the effect of nitroreductase on the reduction of nitro substrates at different temperatures? Also, how did the authors control the reaction temperature in the solar-driven experiment?

Reply: The reaction temperature is specified in the manuscript as rt (room temperature), which is indicated in the reaction schemes of Figures 2-5. To maintain a relatively constant temperature of 25°C in the photoreactor, air refrigeration was incorporated. We observed only a slight variability in reaction temperature ($\pm 3^\circ\text{C}$) depending on the applied light emission. This is now clearly mentioned in the respective sections of the revised Supplementary Information. The solar-driven photoenzymatic synthesis was performed at room temperature in a climate-controlled room. This is now also mentioned in the respective section of the revised Supplementary Information. The reaction performed almost equally well at temperatures between 20-37°C; significantly lower conversions were observed when the reactions were performed at temperatures lower than 20°C or higher than 37°C.

11. In the SI, some compounds have ^{13}C -NMR spectra while some not, please unify the way of supplementing the data.

Reply: All new compounds were fully characterized by ^1H NMR, ^{13}C NMR, and HRMS. The corresponding spectra are included in the SI. For those compounds that have been reported before, we have measured ^1H NMR data and included a statement that the spectral data is in agreement with previously reported spectral data. In a few cases, we also added HRMS and/or ^{13}C NMR data just to further strengthen the product identification, which is especially important for the azobenzenes and when the corresponding literature data is not very convincing. We feel this is a careful and appropriate manner of reporting compound characterization data.

12. In Table S1, what was the condition for the standard enzymatic reaction? Only without LEDs light? The same confusion raised in standard aerobic enzymatic reaction of Table S2.

Reply: The standard enzymatic reactions denote the enzymatic reactions without photocatalyst and LEDs light. For clarity, the precise conditions have now been added to Tables S1 and S2 in the revised Supplementary Information.

13. Was the light intensity of white, blue and red the same?

Reply: Yes, the light intensity of white, blue and red light was the same. The intensity was measured with a lux meter and set up at 1000 lx for all reactions and conditions tested. This is now clearly mentioned in the revised Supplementary Information.

14. Double peaks appeared in the gas chromatography in Figure S31 and S38, are the authors sure that this represents a single compound? My experience with GC/HPLC is that this indicated two compounds.

Reply: MS analysis showed that these seemingly double peaks have the same mass, corresponding to the represented product. Furthermore, this elution pattern was confirmed with a commercial reference compound. Both the product sample and reference compound give the same mass and ¹H NMR data. Hence, we conclude that the data in the gas chromatography in Figure S31 and S38 represent a single compound. This has now also been mentioned in the legends of Figures S31 and S38 of the revised Supplementary Information.

15. By using different starting materials, the reaction mechanism can be proposed and schematically presented.

Reply: This is an excellent suggestion. As already mentioned above in more detail, we have subjected the different starting materials (substrate and reduction intermediates) to the three different catalytic systems (enzyme, photocatalyst, photobiocatalytic system). We have examined the resulting products by GC-MS; the results have been included as Tables S8-S11 in the revised Supplementary Information. Based on these and previous results, a reaction mechanism has been proposed (Scheme S1 and corresponding text in the SI). The Discussion section of the main text has been rephrased accordingly.

16. In page 8 the authors claimed that “we could not observe the reduction of nitrobenzene (1a) into azobenzene (53) with the photoenzymatic system using white LEDs (Suppl. Table S4), We hypothesize this to be the result of insufficient (non-optimal) irradiation of the chlorophyll by the white LEDs”, however, in a latter study the product formation was observed under red LED. To me this is strange as the while LED is often composed of red, green and blue emissions.

Reply: Indeed red light is part of the spectrum of the applied white light (at 1000 lx), but the intensity of this red light is much lower than when applied directly via red LEDs (at 1000 lx). The included time course measurement (Graph S3 in the SI) also nicely shows the accumulation of the desired azoxybenzene intermediate under white light irradiation, which then is efficiently converted into the corresponding azobenzene by applying red light irradiation. Given that red

light is part of the spectrum of white light, the use of increased intensity of white light and increased reaction time could potentially also result in the photoenzymatic conversion of nitrobenzenes into azobenzenes, without the need for a switch from white to red light. However, we have chosen not to use higher light intensities and prolonged incubation times to avoid possible issues with photobleaching and inefficient cooling. This is now clearly stated in the revised manuscript.

17. Though in all cases the conversion and yield were reported, the product concentration should be mentioned to see the robustness, as in a photocatalytic setup (especially when O₂ was supplied), evaporation of the substrate and product could happen.

Reply: All reactions were performed in a capped vial or Erlenmeyer flask to avoid evaporation, using dioxygen saturated buffer without the need for additional supply of O₂. This is now clearly mentioned in the revised SI. Furthermore, reaction progress curves showing product formation in time are now included in the Supplementary Information (Graphs S1–S4 in the revised Supplementary Information).

18. It would be very helpful if a time course could be provided, in such a way the readers could see the kinetic of the reaction process.

Reply: This is an excellent suggestion. Accordingly, we have now included time course measurements, following substrate consumption and product formation (Graphs S1–S4 in the revised Supplementary Information).

19. The SI figures S76, S83, S86 and S88, the proton spectra are not clean, and it is recommended that the authors replace them with clean spectra.

Reply: While these spectra indeed are not fully clean, the products signals are easily recognized. Unfortunately, some of the additional signals are related to the presence of trace amounts of grease, which is difficult to remove. We therefore opted to keep the current spectra.

Reviewer #2 (Remarks to the Author):

The authors set out to fill a gap in the ene reductase (ER)-catalyzed reduction of ene and nitro functionalities: the reduction of aromatic nitro compounds to amines. While ene functionalities and aliphatic nitro compounds have been successfully reduced with photo-assisted ER catalysis, this feat had not been achieved with aromatic nitro compounds. However, while the authors cited Todd Hyster's work (now at Cornell U.), albeit sparsely (a single ref. 42), they did not include Hyster's most recent work, nor the important recent work by Bornadel et al. (Org. Process Res. Dev. 2021, 25, 648–653) and Bisagni et al. (Curr Res Chem Biol, 2022, 100026) from Johnson Matthey and Amgen. This has to be fixed in the revision.

Reply: As suggested by the reviewer, we have now cited some of Hyster's most recent work (refs 48-50), as well as that of Bornadel et al. (ref 31) and Bisagni et al. (ref 32).

That issues aside, the presented work has been performed with care and the results are interesting. The finding of very high chemoselectivity dependent on the oxidant (to azo or azoxy compounds) or the presence of an inert atmosphere (to amines) is highly significant. Likewise for overriding the Nef reaction in the case of aliphatic nitro compounds. After revision, this manuscript should be considered for acceptance.

Reply: we thank the reviewer for these positive comments.

Detailed critique:

Figures 2, 3, and ff: In the Figure legend, clarify the meaning of the numbers next to each product, both before and inside the parentheses: is this conversion and isolated yield?

Reply: As this question might also occur to a reader, we have rephrased the corresponding sentences in the captions of figures 2-5. These sentences now read: “[a] The numbers without brackets denote conversion; the conversion was determined by GC-MS (reaction time = 12 h). [b] The numbers within brackets denote isolated yield; the isolated yield was calculated after workup and purification (reaction time = 24 h).”

Except for the aliphatic amines, the manuscript uses EcNR and BaNTR1 for very different reactions. Is there no cross-reactivity? If so, the reactivity of the ‘other’ enzyme should be clearly explained; if not, it should be clearly mentioned.

Reply: As mentioned in the manuscript, BaNTR1 was not able to perform the reduction of the two representative nitroaliphatics (65a and 66a). Out of all nitroreductases tested, only EcNR was found to accomplish the formation of minor amounts of the respective amine product from both nitro compounds. Hence, EcNR was selected for the photobiocatalytic reaction optimization experiments to achieve efficient conversion of nitroaliphatics into the corresponding aliphatic amines. On the other hand, EcNR exhibits noticeable reductase activity towards nitroaromatics, but this enzyme is less efficient and has a relatively narrow substrate scope compared to BaNTR1. This has now also been clearly mentioned in the revised manuscript.

How did other enzymes on the ‘in-house’ panel (which enzymes does that entail?) do on the aromatic substrates? The manuscript seems to carefully and selectively report positive results (which is great) but should also mention, within the scope, what did not work.

Reply: We would like to emphasize that we also carefully report what did not work. For example, in panel B2 of Figure S2, we have mentioned the nitro compounds that were tested for photoenzymatic conversion but that are not accepted as substrates. We feel this information is important for a reader to understand the scope of the tailored photobiocatalytic systems. Note that the in-house panel of flavin-dependent (nitro)reductases is described in detail in one of our recently published papers (ref 45), which is fittingly cited in the manuscript. While most

of these enzymes display nitroreductase activity towards nitroaromatics, BaNTR1 exhibits the highest activity and broadest substrate scope (unpublished results). However, we feel that a detailed comparison of the substrate scope of all panel members is beyond the scope of this study. Such a detailed study of the substrate scope and catalytic promiscuity of this panel of flavin-dependent (nitro)reductases is ongoing and will be published in due course.

Transformation of 67 to 83: does the nitrocyclohexane also react? How? Selectively? If not, why not?

Reply: We did not test nitrocyclohexane, but the related compound nitrocyclopentane (Figure S2, compound 67b), which is not reactive. The non-reactivity of 67b as well as other nitro compounds (Figure S2, panel B) may result from steric hindrance (enzyme selectivity) and/or a deactivating effect, as mentioned in the manuscript. EcNR seems to have a preference for substrates harboring a phenyl ring (or cyclohexene group), which can either be in conjugation with the nitro functionality or separated from the nitro group by a methylene bridge. This is now also clearly mentioned in the revised manuscript.

Reviewer #3 (Remarks to the Author):

The authors developed a combined photochemical and biocatalytic approach to convert nitro compounds into amines, azoxy- and azo- compounds. The intriguing feature of the study is that the desired reaction route can be selected, based on the reaction conditions. The authors demonstrate the versatility of their system on large library of electronically and structurally distinct aromatic and aliphatic substrates. Unfortunately, no additional information besides the extended substrate scope is provided. From my perspective the reaction mechanism of the distinct reactions (enzymatic and photochemical) should be elucidated. As the reduction proceeds via a range of intermediates (e.g., nitroso, hydroxylamine, imine, oxime), these intermediates should be characterized.

Reply: As suggested by the reviewer, and described in somewhat more detail below, we have included time course measurements for the different photobiocatalytic processes, following substrate depletion and detecting intermediate formation (graphs S1–S4 in the revised Supplementary Information). To clarify the individual contribution of the enzymatic and photochemical system to the overall reaction, we have exposed different starting materials (substrate and observed reduction intermediates or products) to the different catalytic systems (only the enzymatic system, only the photochemical system, and the full photobiocatalytic system). We have examined the resulting products by GC-MS; the results have been included as Tables S8–S11 in the revised Supplementary Information. The outcomes indicate that the photochemical system (chlorophyll) cannot perform the reduction of any substrate or intermediate by itself, but apparently works in concert with the enzyme to reduce the challenging nitroso and azoxy intermediates (of course depending on the applied reaction conditions). In contrast to the nitro and hydroxylamine compounds, the nitroso and azoxy compounds are poorly processed by the individual enzyme system, but are readily converted

by the full photobiocatalytic system under the applied reaction conditions. Based on these new and previous results, a reaction mechanism has been proposed (Scheme S1 and corresponding text in the revised SI). The Discussion section of the main text has been rephrased accordingly.

Overall, the study provides exciting results, but many questions remain open:

- The demonstrated systems work to some degree also without the use of light and also without the use of enzyme:

O The reduction to the amine works with the ruthenium based photocatalyst without enzyme. I wonder whether chlorophyll and red light, or using chlorophyll with a higher light intensity would also show some reactivity.

Reply: Photochemical reduction to the amine is not observed with chlorophyll and red or white light at the applied intensity (1000 lx). Photochemical background reactivity is observed with blue light (1000 lx). The control reactions reported in Table S2 suggest that the observed reduction of nitrobenzene to aniline in the absence of enzyme largely results from the photochemical activity of the photoexcited NADH cofactor under blue light irradiation, as previously reported in the literature (appropriate citations are given in the manuscript). Indeed, no photochemical reduction was observed when removing components of the nicotinamide recycling system (glucose, NAD⁺ or glucose dehydrogenase). Consistently, photochemical conversion was observed in the presence of 20 mM NADH (Table S2). We feel that this is clearly mentioned in the manuscript. We have not tested higher light intensities, or prolonged incubation times, to avoid possible issues with photobleaching and inefficient cooling. This is now clearly stated in the revised Supplementary Information.

The reaction works also without the photocatalyst. This suggests to me that the photocatalyst and the enzyme do have different reaction rates for the different reduction intermediates and in concert both can play their strengths. However, in order to elucidate this, the controls need to be made with the potential intermediates (nitroso, hydroxylamine, imine, oxime) using the full system, only the enzymatic system and only the photochemical system. This should provide more data on the individual catalysts` contribution to the overall reaction.

Reply: This is an excellent suggestion. Accordingly, we have included time course measurements for all four different photobiocatalytic processes, following substrate consumption and examining intermediate/product formation (graphs S1–S4 in the revised Supplementary Information). In addition, we have exposed different starting materials (substrate and reduction intermediates) to the different catalytic systems (only the enzymatic system, only the photocatalytic system, and the full photobiocatalytic system) and examined the corresponding intermediates/products by GC-MS. The results have been included as Tables S8-S11 in the revised Supplementary Information and indicate that the photochemical system (chlorophyll) cannot perform the reduction of any substrate or intermediate by itself, but apparently works in concert with the enzyme to reduce the challenging nitroso and azoxy intermediates (of course depending on the applied reaction conditions). Indeed, in contrast to the nitro and hydroxylamine compounds, the nitroso and azoxy compounds are poorly

processed by the individual enzyme system; however, they are readily converted by the full photobiocatalytic system under the applied reaction conditions. This is now included in the revised manuscript and Supplementary Information.

o The formation of the azoxy-compounds would work without applying the photochemical system. Similar as in the previous point, the reactivities of the intermediates should be characterized.

Reply: This has been done for all transformations, as explained in the previous response. See Tables S8-S11 in the revised Supplementary Information.

o The conversion of the aliphatic compounds requires the addition of ascorbate, which again, on its own would perform the reaction. Again, the two catalytic systems should be tested with all potential reaction intermediates.

Reply: Again, we have included a time course measurement and tested the different catalytic systems with different starting materials (substrate and the reduction intermediates). The results are presented in Graph S4 and Table S11 and indicate that the photochemical system and ascorbic acid contribute to the reduction of the nitroso intermediate, avoiding the tautomerization pathway to the oxime. The enzymatic system seems largely responsible for the other two reduction steps, elucidating how the different catalytic systems work together. This is now clearly mentioned in the revised manuscript.

-All this information should be collected in order to propose a reaction sequence with the individual intermediates. Only then the role of the photochemical and the biocatalyst is clear.

Reply: Based on the new and previous results, a reaction mechanism has been proposed (Scheme S1 and corresponding text in the revised manuscript and Supplementary Information). The individual enzyme readily reduces the nitro and hydroxylamine compounds, most likely by hydride transfer (and accompanied by proton acquisition from solvent). Given that the nitroso and azoxy intermediates are poorly processed by the individual enzyme system, but are readily reduced by the full photobiocatalytic system under the applied reaction conditions (Tables S8-S11), we propose that the photocatalytic assistance is executed at these steps. The results support the hypothesis that chlorophyll works in synergy with the enzyme, most likely by promoting the transfer of a single electron (SET) to the nitroso and azoxybenzene intermediates, accompanied by a hydrogen atom transfer (HAT) by the flavoenzyme. For each step in the reaction sequence (hydride transfer or SET/HAT), both electrons thus likely result from the reduced nicotinamide cofactor and are shuttled via the flavin cofactor (see scheme S1 in the revised SI).

-The formation of the azoxy-compounds requires a switch of light. This is surprising, as red light should be part of the spectrum of the applied white light. Is the reason the different light intensity? Then either increased reaction time or increased intensity of white light should give the same effect. This effect should be explained.

Reply: The reviewer is absolutely correct. Red light is part of the spectrum of the applied white light (at 1000 lx), but the intensity of this red light is indeed much lower than when applied directly via red LEDs (at 1000 lx). The included time course measurement (Graph S3 in the SI) also nicely shows the accumulation of the desired azoxybenzene intermediate under white light irradiation, which then is efficiently converted into the corresponding azobenzene by applying red light irradiation. Given that red light is part of the spectrum of white light, the use of increased intensity of white light and increased reaction time could potentially also result in the photoenzymatic conversion of nitrobenzenes into azobenzenes, without the need for a switch from white to red light. However, we have chosen not to use higher light intensities and prolonged incubation times to avoid possible issues with photobleaching and inefficient cooling. This is now clearly stated in the revised Supplementary Information.

-The reaction conditions of the photochemical reactions are not sufficiently documented. What are the different applied light intensities and emission spectra? This data should be available from the manufacturer of the LED. It is my understanding that the authors utilized a custom illumination system. In this case the distance between sample and light source should be provided. Also, the light intensity in the reactor should be put into context with the light intensity that was available in the reported solar-powered reactions. Finally, the absorption spectrum of the reaction mixture and the photocatalyst should be provided in order to discuss the requirement of two different light sources for the formation of the azo compounds.

Reply: The photoreactors employed were designed in-house, using white, blue (λ max 440 nm) or red (λ max 660 nm) LED stripes (19 W, 1000 lx) coupled with air refrigeration. The light intensity of the applied white, blue and red light was the same. The intensity was measured with a lux meter and set up at 1000 lx for all reactions and conditions tested. The chlorophyll employed was obtained from TCI Chemicals (C0780), with a λ max at 660 to 670 nm (diethyl ether). The capped glass vial containing the reaction mixture was placed in a circular disposition at a distance of 6 cm from the light source. All this information is now clearly mentioned in the revised Supplementary Information. The requirement of two different light sources for the formation of the azo compounds has been addressed in our reply to the preceding question. For the solar-powered reactions, sunlight intensity was reported in W/m^2 and there is no simple conversion to lx, which depends not only on the wavelength (or color) of the light, but also on meteorological conditions, light angle, refraction of objects, day time, etc. For clarity, we have rephrased the corresponding sentence in the revised manuscript, which now reads: "A typical scale-up reaction was carried out under sunlight exposure for 48 h, bearing in mind the variability in light intensity produced by the periods of night, dawn and dusk, as well as other meteorological phenomena." Note that we purely wanted to demonstrate that the use of chlorophyll as photocatalyst enables the use of sunlight to drive the tailored photobiocatalytic reactions.

-The contribution of the different electron donors should be unraveled by control experiments. The applied buffer compound MOPS was demonstrated to serve as electron donor in photochemistry. In addition, a glucose based NADH regeneration system is applied. MOPS

might promote the photochemical reaction, glucose the biocatalytic one. Alternate buffer salts and water should be tested as solvents. The number of NADH equivalents that are utilized in the reaction should be measured and correlated to the number of electrons that are required for the reduction. And the two processes (photochemical and biocatalytic) should be tested individually with the different possible electron donors.

Reply: This is an excellent suggestion. We have tested alternative inorganic buffers (NaPi, NaHCO₃ and KH₂PO₄), and found that these inorganic buffers can also support the photobiocatalytic process, albeit resulting in somewhat lower conversions (62-77% conversion of nitrobenzene into aniline) compared to the transformations in MOPS. This is now clearly mentioned in the respective section of the revised Supplementary Information. On the other hand, the photobiocatalytic transformations are fully dependent on NADH, supplied via a glucose-based NADH regeneration system (see Tables S2-S4 and S7). In the presence of both MOPS and the glucose-based NADH regeneration system, the individually tested photocatalytic system (in contrast to the individually tested nitroreductase) did not show any reactivity towards the starting substrate or intermediates (Tables S8-S11 in the revised Supplementary Information). Hence, chlorophyll itself, without the presence of the nitroreductase, cannot perform the reduction of any substrate or intermediate. The results support the hypothesis that chlorophyll works in synergy with the enzyme, most like by promoting the transfer of a single electron (SET) to the nitroso and azoxybenzene intermediates (which appear to be relatively poor substrates for the individual biocatalytic system – see Tables S8-S11), with the flavoenzyme likely performing a hydrogen atom transfer (HAT). For each step in the reaction sequence (enzymatic hydride transfer or photobiocatalytic SET/HAT), both electrons thus likely result from the reduced nicotinamide cofactor and are likely shuttled via FMN (see proposed mechanism in scheme S1 in the revised SI). Note that the complex reaction mixture makes the measuring of the number of NADH equivalents utilized by UV-VIS spectroscopy basically impossible.

-Page 6: Figure 2 and the following line: “Furthermore, various dinitrobenzenes and nitroanilines can be fully converted into the benzenediamine products (24-30 and 31).” Substrates 16 and 17 also belong to this group.

Reply: The reviewer is correct. Accordingly, this has been adjusted in the revised manuscript.

-Caption of figure 2: I believe in point [e] the wrong substrate number is given. Maybe 1g instead?

Reply: The reviewer is correct. Accordingly, this has been corrected in the revised manuscript.

-The captions of all figures: the meaning of [a] and [b] is not obvious. I suggest writing “numbers in brackets” and “numbers without brackets”

Reply: This is an excellent suggestion. Accordingly, we have rephrased the corresponding sentences in the captions of figures 2-5. These sentences now read: “[a] The numbers without

brackets denote conversion; the conversion was determined by GC-MS (reaction time = 12 h). [b] The numbers within brackets denote isolated yield; the isolated yield was calculated after workup and purification (reaction time = 24 h)."

-Table S6: reactions performed on "bench": does this mean without light? This should be clarified in the text.

Reply: The reactions were performed on bench under ambient light. This is now clearly mentioned in the revised Supplementary Information.

-It would be great to see an aliphatic substrate that might give an e.e. to demonstrate the enzyme's stereoselectivity.

Reply: In this study we focused on the development of photoenzymatic systems for the selective synthesis of various aliphatic amines and amino-, azoxy- and azo-aromatics from the corresponding nitro compounds. For the aliphatic substrates, we have concentrated on those that yield terminal amines, which are well accepted by the enzyme. Although we agree with the reviewer that it would be interesting to investigate other aliphatic substrates to produce enantioenriched amines, we feel that such an investigation into the enzyme's stereoselectivity requires the careful testing of a wide range of structurally diverse aliphatic substrates, which is really beyond the scope of this study and one of the topics of our future work.

REVIEWER COMMENTS

Reviewer #1 (Remarks to the Author):

The authors have adequately addressed my concerns in the first round. Though the manuscript could be further strengthened by the elucidation of the reaction mechanisms, I feel that at this stage it can be accepted for publication as is in Nature Communications.

Reviewer #2 (Remarks to the Author):

In their revision, the authors addressed all points that this reviewer deems relevant and necessary for a decision to accept the manuscript for publication. The authors have satisfactorily addressed all outstanding issues.

Reviewer #3 (Remarks to the Author):

The authors have addressed the majority of all three reviewer's comments and the overall quality of the manuscript has improved significantly. It is close to state where I would recommend its publication in Nature Communications. However, one important issue remains that must be resolved before it can be published in any journal.

The reaction engineering and mechanistic discussions of the authors rely significantly light intensities measured in lux. However, the photochemically relevant unit should be the photon flux (in photons/second and per volume, e.g., in Einstein/volume) and not lux.

Using lux has two major issues and is scientifically wrong:

- Lux is derived from lumen and lumen is related to the light energy, but arbitrarily scaled to the way the human eye perceives light intensity at different wavelengths. This means the factor for converting the light intensity from lumen to its energy is different at every wavelength. It is especially low at the regions of the visible spectrum that are not well recognized by the human eye (e.g. red and blue light).
- Using a unit of energy is also not recommended in photochemistry, as light at different wavelengths has different energy. In other words: the same number of photons have double the energy at 400 nm as they would have at 800 nm. The important measure for a photochemical reaction is therefore the number of absorbed photons and not their energy. I understand that from an energy-perspective also units such as Joule or Watt are important, but comparing the performance of photochemical reactions and drawing scientific conclusions from this comparison is not possible using energy as a parameter. This requires knowledge about the photon flux and the number of absorbed photons.

The authors must either accurately measure the photon flux inside the reaction vessel at the respective conditions using chemical actinometry (which I recommend) or provide a rough estimation of the photon flux from a correct conversion of lux to photon flux using the appropriate factors at the respective wavelength. Otherwise, all conclusions that are drawn based on light intensity must be removed from the manuscript.

Furthermore, in my comments I suggested that the authors provide a UV-Vis spectrum of a representative reaction mixture. I still suggest that the authors provide this data, as this allows to estimate the fraction of photons that are absorbed from the photon flux.

I still support publication of the manuscript in Nature Communications, however, this issues must be fixed.

Response to the reviewers' comments

We appreciate the remarks of the reviewers as well as the decision to consider a revised manuscript, which addresses the points raised by Reviewer 3. The reviewer's comments are provided below along with our responses (in italics). The changes in the text are highlighted by track changes in the revised Manuscript and Supplementary Information.

Reviewer #1 (Remarks to the Author):

The authors have adequately addressed my concerns in the first round. Though the manuscript could be further strengthened by the elucidation of the reaction mechanisms, I feel that at this stage it can be accepted for publication as is in Nature Communications.

Reply: we thank the reviewer for these positive comments.

Reviewer #2 (Remarks to the Author):

In their revision, the authors addressed all points that this reviewer deems relevant and necessary for a decision to accept the manuscript for publication. The authors have satisfactorily addressed all outstanding issues.

Reply: we thank the reviewer for these positive comments.

Reviewer #3 (Remarks to the Author):

The authors have addressed the majority of all three reviewer's comments and the overall quality of the manuscript has improved significantly. It is close to state where I would recommend its publication in Nature Communications.

Reply: we thank the reviewer for these positive comments.

However, one important issue remains that must be resolved before it can be published in any journal. The reaction engineering and mechanistic discussions of the authors rely significantly on light intensities measured in lux. However, the photochemically relevant unit should be the photon flux (in photons/second and per volume, e.g., in Einstein/volume) and not lux.

Using lux has two major issues and is scientifically wrong:

- Lux is derived from lumen and lumen is related to the light energy, but arbitrarily scaled to the way the human eye perceives light intensity at different wavelengths. This means the factor for converting the light intensity from lumen to its energy is different at every wavelength. It is especially low at the regions of the visible spectrum that are not well recognized by the human eye (e.g. red and blue light).
- Using a unit of energy is also not recommended in photochemistry, as light at different wavelengths has different energy. In other words: the same number of photons have double the

energy at 400 nm as they would have at 800 nm. The important measure for a photochemical reaction is therefore the number of absorbed photons and not their energy. I understand that from an energy-perspective also units such as Joule or Watt are important, but comparing the performance of photochemical reactions and drawing scientific conclusions from this comparison is not possible using energy as a parameter. This requires knowledge about the photon flux and the number of absorbed photons.

Reply: Note that in order to ascertain the reproducibility of the experimental data, we report the exact distance between the light source and the reaction vial, where the light intensity was precisely measured with a lux meter and set up at 1000 lx for all reactions and conditions tested. For the solar-powered reactions, sunlight intensity was reported in W/m^2 . However, the reviewer is correct in stating that an accurate comparison of the performance of photochemical reactions requires precise information about the photon flux and the number of absorbed photons.

The authors must either accurately measure the photon flux inside the reaction vessel at the respective conditions using chemical actinometry (which I recommend) or provide a rough estimation of the photon flux from a correct conversion of lux to photon flux using the appropriate factors at the respective wavelength. Otherwise, all conclusions that are drawn based on light intensity must be removed from the manuscript.

Reply: It is not really possible to measure the photon flux for white light, and for red light (680 nm) it is quite difficult since chemical actinometers for the reliable characterization of reactions driven by visible light above 600 nm seem to be largely missing (Photochemistry and Photobiology, 2021, 97: 873–902). Unfortunately, there is no simple conversion of lux into photon flux, and we do not feel comfortable including numbers for photon flux that are very rough estimations based on several assumptions rather than accurate numbers. Therefore, as suggested by the reviewer, we have chosen to delete or rephrase several sentences in the revised manuscript and revised supplementary information to take out conclusions that are drawn based on light intensity. We also strongly feel that these sentences are not really needed, since we merely aimed to establish effective conditions for the desired photobiocatalytic transformations.

Furthermore, in my comments I suggested that the authors provide a UV-Vis spectrum of a representative reaction mixture. I still suggest that the authors provide this data, as this allows to estimate the fraction of photons that are absorbed from the photon flux.

Reply: We have now included a UV-VIS spectrum of a representative reaction mixture as Graph S5 in the Revised Supplementary Information. The reaction conditions are the same as employed for the reduction of nitrobenzenes into anilines. The substrate used was 4-nitrobenzonitrile, one of the best performing substrates for all reported reactions.

I still support publication of the manuscript in Nature Communications, however, this issues must be fixed.

Reply: As suggested by the reviewer, we have chosen to rephrase several sentences to remove all conclusions that are directly based on light intensity from the manuscript. We hope that the manuscript is now suitable for publication.

REVIEWERS' COMMENTS

Reviewer #3 (Remarks to the Author):

The authors have adequately addressed all reviewers comments and I support the publication of the manuscript in its current state.